# Can the Healthy Start Risk Screen Predict Perinatal Depressive Symptoms among High-Risk Women?

**DOI:** 10.3390/children9020180

**Published:** 2022-02-01

**Authors:** Roneé Wilson, Adriana Campos, Mannat Sandhu, Sarah Sniffen, Rashida Jones, Hope Tackett, Estrellita Berry, Adetola Louis-Jacques

**Affiliations:** 1College of Public Health, University of South Florida, Tampa, FL 33612, USA; elba@usf.edu (A.C.); rashidajones@usf.edu (R.J.); 2USF Health Morsani College of Medicine, University of South Florida, Tampa, FL 33612, USA; msandhu@usf.edu (M.S.); ssniffen@usf.edu (S.S.); 3REACHUP, Incorporated, Tampa, FL 33607, USA; htackett@reachupincorporated.org (H.T.); lberry@reachupincorporated.org (E.B.); 4College of Medicine, University of Florida, P.O. Box 100294, Gainesville, FL 32610, USA; louisjacquesa@ufl.edu

**Keywords:** healthy start, health disparities, perinatal depression, maternal health

## Abstract

Objectives: Early detection of depression in at-risk populations is critical for ensuring better maternal and child health outcomes. This study assessed whether Healthy Start Prenatal Risk Screening (HSPRS) could predict depressive symptoms in women enrolled in a Healthy Start (HS) program in under-resourced, high-risk communities of Hillsborough County. Methods: Data from HS participants were included for those who were evaluated using the HSPRS and the Edinburgh Postnatal Depression Scale (EPDS). A correlation analysis determined if the HSPRS score was associated with a positive EPDS screen, and HSPRS questions related to the participants psychosocial environment were assessed individually to determine their predictive potential. The crude odds ratio (OR) and adjusted OR (controlling for sociodemographic covariates) were calculated for each question of interest. Results: A total of 736 women were included, with 122 (16.5%) scoring 14 or greater on the EPDS, indicating probable depression risk. There were significant differences between women at risk for depression compared to those not at risk regarding maternal age (*p*-value = 0.03) and marital status (*p*-value = 0.01). There were no significant differences in education, ethnicity, or race. The total HSPRS score had a weak yet significant correlation with the EPDS score (r = 0.14, *p*-value = 0.0001), and seven individual HSPRS questions were significantly associated with risk for perinatal depression. Conclusions for Practice: By focusing on responses to key HSPRS questions rather than the overall score, women may receive access to much needed services more quickly, thereby reducing the risk for poorer maternal and developmental outcomes. Significance: A young maternal age and single marital status have been identified as risk factors for perinatal depression. Additionally, women from racial/ethnic minority groups or low-income populations are more likely to experience depression. Thus, in communities where women exhibit many pre-identified risk factors for perinatal depression, the ability to quickly identify those at the highest risk is imperative. This work indicates that among medically and socially high-risk mothers enrolled in a HS program, the overall HSPRS score was not as predictive of perinatal depression as individual responses to key questions. Attention to these responses could result in women receiving much needed services quicker.

## 1. Introduction

Perinatal depression, defined as maternal depression during pregnancy and up to one year after giving birth, is associated with negative maternal and infant outcomes [1,2,3,4,5]. A depressive episode affects about 19% of postpartum women, with most episodes occurring during the first 3 months postpartum [2]. Depressed mothers are less likely to seek prenatal care and are more likely to experience adverse perinatal outcomes including delivering prematurely (<37 weeks gestation) or giving birth to a low birth weight infant (<2500 g) [3,6]. Infants born to depressed mothers may be more likely to experience altered immune system functioning, less healthy sleep practices, reduced breastfeeding initiation, and infant mortality [3,7,8]. In extreme cases, perinatal depression can result in maternal suicide, which is the second leading cause of death among postpartum women [9].

Women from racial/ethnic minoritized groups or low-income populations are more likely to experience depression compared to the majority group or to their higher socioeconomic status counterparts [2,10]. Additionally, single relationship status; substance use; and a history of trauma, including domestic violence, physical abuse, and/or sexual abuse, are associated with higher risk of depression during the perinatal period. Importantly, women with lower incomes are more likely to experience these risk factors. Specifically, lower income populations are at higher risk for exposure to trauma, which is associated with poor health behaviors during pregnancy and adverse infant outcomes [11]. Thus, it is critical that particular attention is paid to perinatal mental health screening among at-risk mothers. 

Among the women already identified as high-risk based on demographic characteristics, it is especially important to assess other social determinants of health using multiple evaluation tools, as these tools may be instrumental in identifying mothers who are at increased risk of experiencing or developing perinatal depression. Considering the importance of early detection to improving health outcomes in at-risk populations, this study aimed to assess if the Healthy Start Prenatal Risk Screen (HSPRS) could be used to predict depressive symptoms in women enrolled in an urban HS program.

## 2. Materials and Methods

### 2.1. Setting

All pregnant women in the state of Florida are offered the HSPRS, a series of questions designed to determine whether the mother may benefit from the program. The screening instrument gathers self-reported information on demographic characteristics, as well as environmental and social factors, which may increase the risk for adverse birth outcomes [1]. The Central Hillsborough Healthy Start (CHHS) program is a federally funded program managed by REACHUP, Incorporated, which provides family-centered preconception, prenatal, and interconception risk reduction services to under-resourced and high-risk communities of Hillsborough County, Florida. Participation in CHHS is completely voluntary and all services are provided free of charge. Importantly, the risk reduction services include screening for perinatal depression, a critical metric for CHHS’s service area where adverse perinatal outcomes such as infant mortality rates are at least 1.5 times the U.S. national average. Three-quarters of CHHS’s community identify as a racial/ethnic minority, and the majority of births are to black mothers who are typically young, unmarried, and Medicaid-eligible. Compared to the rest of the county, families in the CHHS project area tend to be poorer, earning half the median county income and experiencing double the unemployment rate. These are all literature-identified risk factors that have been shown to increase the risk of perinatal depression.

### 2.2. Study Materials

HSPRS is the initial assessment all pregnant women receive and it allows CHHS providers to identify those who may be vulnerable to adverse health outcomes [12]. The form contains a total of 16 questions, both demographic and behavioral [12]. Each affirmative answer is worth one point, except for race = Black, which is worth two points [12], adding additional emphasis on the racism and discrimination often experienced by minoritized groups. The total number of points scored are summed together to provide a total HSPRS score. According to the Florida Department of Health, women who score a four or higher on the HSPRS are six times more likely to experience post-neonatal infant mortality compared to women who score less than four [12]. 

The Edinburgh Postnatal Depression Scale (EPDS) is a 10-item self-reporting questionnaire that has been used extensively and is validated across cultures as an accurate depression screening tool [8,13,14]. The scale can be used to screen for depressive symptoms in pregnant and postpartum women, and provides a satisfactory sensitivity and specificity [13,14,15,16,17]. Response items are scored from 0 to 3 to account for increased symptom severity. A score of 14 or higher represents a positive screen and is indicative of probable depression [15,16].

Women who are ultimately enrolled in the CHHS program typically receive HSPRS at their medical provider’s office approximately 6–9 weeks prior to initiating Healthy Start services. Thus, there is a considerable gap in the time between completing HSPRS and receiving an EPDS screen at the first Healthy Start visit. 

### 2.3. Participants

Data from CHHS participants who completed both the HSPRS and EPDS between 2009 to 2015 were included in the study. If a participant had multiple screenings during the study period, the first EPDS score was used in the analysis. This study was approved by the Institutional Review Board at the University of South Florida. Participants provided consent for their data to be used for research and evaluation (Pro00001724). 

### 2.4. Study Variables

The primary exposure variables were total HSPRS score and individual psychosocial risk screening questions (i.e., questions 1 through 10). An EPDS score of 14 or greater was considered indicative of perinatal depression. Correlation analysis was used to determine whether the total HSPRS score or individual items have could predict perinatal depression (Figure 1). Participant sociodemographic characteristics were assessed as covariates. These included maternal age, marital status (“married” or “single”), education (“less than high school”, “at least high school”, “post high school”, or “unknown”), race (“White”, “Black”, or “unknown/other”), and ethnicity (“Hispanic/Latino,” “not Hispanic/not Latino”, or “unknown/not recorded”). The final models included participant age as a continuous variable. 

### 2.5. Statistical Analysis

Demographic variables were analyzed using the Chi-square test if categorical and t-test if continuous. Pearson’s correlation coefficient was calculated to assess the relationship between the total HSPRS score and the total EPDS score. Point-biserial correlation was used to assess the relationship between individual psychosocial environment items on HSPRS, categorized as 0 or 1, and the total EPDS score. The crude odds ratios (OR) and adjusted odds ratio (aOR) were calculated for each question of interest using a positive depression screen (EPDS ≥ 14) using SAS 9.4, and a *p*-value of <0.05 was considered significant. Adjusted odds ratios were calculated using logistic regression controlling for age, education, marital status, race, and ethnicity.

## 3. Results

Of the 736 women included in the study, 122 (16.5%) had a positive EPDS score (≥14) indicating an increased risk of depression. The majority of women in the sample identified as Black/African American and non-Hispanic (Table 1). Most women had a high school education, were unmarried, and were insured during their pregnancy. Women who screened positive on the EPDS were more likely to report they were single (*p*-value = 0.0124). 

The total HSPRS score had a weak, yet significant, correlation with the EPDS score (r = 0.14, *p*-value = 0.0001). Seven of the ten HSPRS questions significantly correlated either positively or negatively with a positive EPDS score. Table 2 depicts the adjusted and unadjusted association between the 10 psychosocial environmental HS screening questions and a subsequent positive EPDS screen. Responding “yes” to the question regarding the presence of children at home with medical or special needs (Q4) indicated the strongest protective measure of association (aOR = 0.225, 95% CI 0.069–0.734), and answering “yes” to the question whether the timing of the pregnancy was good (Q5) was also protective with an aOR of 0.555 (95% CI 0.364–0.845). Answering “yes” to the remaining questions regarding feeling down or depressed in the last month (Q6) (aOR = 3.614, 95% CI 2.401–5.440), feeling alone in the past month (Q7) (aOR = 3.906, 95% CI 2.555–5.971), previously receiving mental health services (Q8) (aOR = 2.250, 95% CI 1.415–3.575), experiencing intimate partner violence in the past year (Q9) (aOR = 2.745, 95% CI 1.400–5.381), and whether the mother has trouble paying the bills (Q10) (aOR = 2.135, 95% CI 1.409–3.236) were associated with an increased risk for scoring positive on the EPDS. Notably, the question that asked whether the mother felt alone when facing problems in the past month (Q7) had the strongest association with a positive EPDS screen. 

## 4. Discussion

HS programs serving participants who are already high-risk for adverse perinatal outcomes, including maternal depression (i.e., single, young mothers, racial/ethnic minorities, and low education attainment) may benefit from exploring alternative methods to identify mothers at risk for depression [1,10,11,18]. Our analysis, similar to the findings of others [19], determined that the timing of pregnancy was significantly associated with depression, as participants answering “yes” to the question regarding whether pregnancy timing was good were less likely to screen positive for depression. Having children at home with medical or special needs (Q4) seemed to have the strongest protective effect, as these mothers were nearly 80% less likely to screen positive for depressive symptoms. There may be multiple reasons for this protective effect, including already having access to perinatal resources, an established support system [20], or feeling more prepared for the current pregnancy [21]. Conversely, multiple studies have implicated “wrong timing” as a risk factor in perinatal depression [19,22,23]. 

Five additional questions (Q6–Q10) were highly associated and significant. Questions focusing on current mental health status, social isolation, previous use of mental health services, and intimate partner violence (Q6–Q9) could be used to predict a positive EPDS score. Similarly, having financial difficulties was associated with a positive score. The two questions (Q6 and Q7) that directly inquired about feelings and/or symptoms commonly associated with perinatal depression were highly associated with a positive EPDS score. Participants who answered yes to feeling depressed/down/hopeless in the last month (Q6) showed the second highest association between answering yes and screening positive on EPDS. The strongest predictor pertained to whether the participant felt alone when facing problems in the past month (Q7). In contrast, the question with the lowest adjusted odds ratio asked about whether the participant had trouble paying her bills (Q10). However, participants who answered yes to having trouble paying their bills were still twice as likely to score a positive EPDS screen compared to those who answered no. 

All questions with a positive association between answering yes and a positive EPDS screen related to established perinatal depression risk factors, including low socioeconomic status [18,24], social isolation [25,26,27], feeling depressed [18], physical abuse [1], and previously using mental health services/counseling [28]. This corroboration further supported the validity of the study results. Additionally, this study aligns with recent work conducted in another Healthy Start cohort, which focused on the correlation between HSPRS and EPDS during the postpartum period [29]. 

The data indicate using HSPRS is a practical alternative screening tool for HS programs whose participants are at high-risk for perinatal depression. However, it should be noted that there are limitations to these analyses, including that the existing data did not allow the researchers to control for multiple factors also associated with perinatal depression, such as childhood trauma or self-esteem [2,11,28,30]. However, the strengths of the study include its high participant number and consistency in the administration of the screening tools. 

## 5. Conclusions

This study confirmed the association between young maternal age, single marital status, and perinatal depression in a cohort of women from a medically and socially high-risk population in Hillsborough County. Additionally, the data indicate that seven of the ten psychosocial environment screening questions included in the HSPRS could be used to quickly identify women who are likely to score positively on the EPDS. Those who score positively on the EPDS merit further evaluation for depression, which is known to have adverse effects on maternal and child health. This is especially important for screening women from racial/ethnic minoritized groups or low-income populations, as they are already at an increased risk for experiencing perinatal depression. The overall HSPRS score was not as predictive of perinatal depression as individual responses to key questions. Focusing on these responses may allow women to receive access to much needed services more quickly, thereby improving the medical and psychological outcomes of mothers and their children. 

## Figures and Tables

**Figure 1 children-09-00180-f001:**
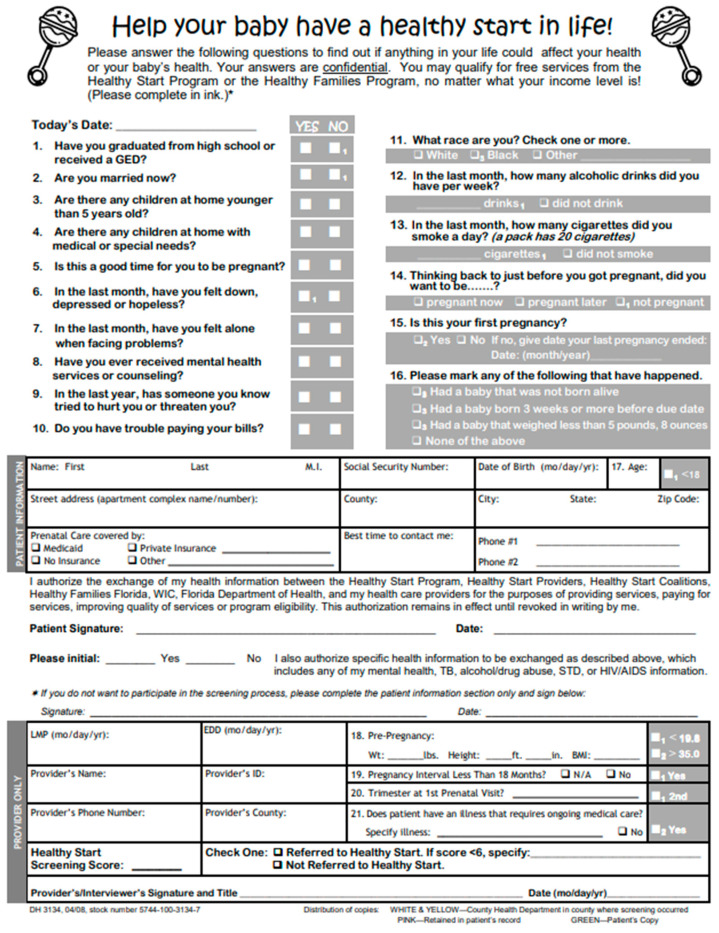
Prenatal screen.

**Table 1 children-09-00180-t001:** Characteristics of Healthy Start participants, Tampa, Florida, 2009–2015.

Variable		Positive EPDS Score n = 122 (16.6%)	Negative EPDS Score n = 614 (83.4%)	*p*-Value
Age	Less than 18	0 (0.00)	27 (4.4)	0.081
	18 to 24	60 (49.2)	256 (42.1)	
	25 to 34	53 (43.4)	279 (45.9)	
	35+	9 (7.4)	46 (7.6)	
Race	White/Caucasian	27 (22.1)	149 (24.3)	0.0797
	Black/African American	85 (69.7)	372 (60.6)	
	Unknown/Other	10 (8.2)	93 (15.2)	
Ethnicity	Hispanic	21 (17.2)	146 (23.8)	0.2799
	Non-Hispanic	73 (59.8)	343 (55.9)	
	Unknown	28 (23)	125 (20.4)	
Education	Less than HS	47 (38.5)	211 (34.4)	0.679
	HS/GED or post HS education	68 (55.7)	365 (59.5)	
	Unknown	7 (5.7)	38 (6.2)	
Marital Status	Single	108 (88.5)	483 (78.7)	0.0124
	Married	14 (11.5)	131 (21.3)	
Insurance	Medicaid/Other Insurance	115 (96.6)	543 (95.9)	0.1933
	No Insurance	4 (3.4)	23 (4.1)	

*p*-values are from a Chi-square test/Fisher exact for categorical variables. Total values for categorical variables may be incomplete due to missing values.

**Table 2 children-09-00180-t002:** Crude and Adjusted Analysis of the Association between Healthy Start Screening Questions and Positive EPDS Screen.

Questions		Crude OR (CI)	Adjusted OR (CI)
Q1	Have you graduated from high school or received a GED?	0.777 (0.520, 1.159)	0.725 (0.412, 1.277)
Q2	Are you married now?	0.653 (0.386, 1.104)	1.287 (0.564, 2.938)
Q3	Are there any children at home younger than 5 years old?	1.040 (0.705, 1.534)	1.024 (0.689, 1.522)
Q4	Are there any children at home with medical or special needs?	0.222 * (0.068, 0.718)	0.225 * (0.069, 0.734)
Q5	Is this a good time for you to be pregnant?	0.523 * (0.347, 0.787)	0.555 * (0.364, 0.845)
Q6	In the last month, have you felt down, depressed or hopeless?	3.627 * (2.427, 5.419)	3.614 * (2.401, 5.440)
Q7	In the last month, have you felt alone when facing problems?	3.972 * (2.625, 6.008)	3.906 * (2.555, 5.971)
Q8	Have you ever received mental health services or counseling?	2.155 * (1.388, 3.348)	2.250 * (1.415, 3.575)
Q9	In the last year, has someone you know tried to hurt or threaten you?	2.736 * (2.736, 5.259)	2.745 * (1.400, 5.381)
Q10	Do you have trouble paying your bills?	2.086 * (1.397, 3.115)	2.135 * (1.409, 3.236)

Model adjusted for age (Units = 5), education, marital status, race, and ethnicity. EPDS = Edinburgh Postnatal Depression Scale. exposure = “yes” and outcome = positive EPDS screen. * Indicates a *p*-value < 0.05.

## Data Availability

The data are not publicly available due to privacy considerations.

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
