# Peer review of "Can the Healthy Start Risk Screen Predict Perinatal Depressive Symptoms among High-Risk Women?"

_children, 2022, doi:10.3390/children9020180_

Round 1

Reviewer 1 Report

It is a remarkable paper about the prediction of prerinatal depressive symptoms among high-risk women. However, several revisions are required as follows:

  1. The potential influence of confounding factors (i.e., marital status) on the findings should be adjusted.
  2.  Significance for p-value should be definitely described in statistical analysis.
  3. Limitations should be additionally described in discussion.

Author Response

We thank the reviewer for the comments. Please see the attachment for author responses.

Reviewer 2 Report

The work is very interesting but requires minor corrections. It is necessary to enter diagnostic criteria for postpartum depression. 

1.The authors should expand on all abbreviations used in the work, e.g. there is no EPDS abbreviation. 

2.The EPDS scale is recommended for use in screening for postpartum depression. 

3.The authors should supplement the article with relevant literature on the recommendation of the scale for depression in pregnancy.

Author Response

We thank the reviewer for the comments. Please see the attachment for the authors' response. 
